# Wound-Microenvironment Engineering through Advanced-Dressing Bioprinting

**DOI:** 10.3390/ijms23052836

**Published:** 2022-03-04

**Authors:** Cristina Del Amo, Xabier Fernández-San Argimiro, María Cascajo-Castresana, Arantza Perez-Valle, Iratxe Madarieta, Beatriz Olalde, Isabel Andia

**Affiliations:** 1Regenerative Therapies, Bioprinting Laboratory, Biocruces Bizkaia Health Research Institute, Cruces University Hospital, 48903 Barakaldo, Spain; cristina.delamomateos@osakidetza.eus (C.D.A.); arantzaperez6@gmail.com (A.P.-V.); 2TECNALIA, Basque Research and Technology Alliance (BRTA), 20009 Donostia-San Sebastian, Spain; xabier.fernandez@tecnalia.com (X.F.-S.A.); maria.cascajo@tecnalia.com (M.C.-C.); iratxe.madarieta@tecnalia.com (I.M.); beatriz.olalde@tecnalia.com (B.O.)

**Keywords:** bioink, 3D bioprinting, decellularized adipose extracellular matrix, plasma, platelet, cytokines, growth factors, wound healing

## Abstract

In patients with comorbidities, a large number of wounds become chronic, representing an overwhelming economic burden for healthcare systems. Engineering the microenvironment is a paramount trend to activate cells and burst-healing mechanisms. The extrusion bioprinting of advanced dressings was performed with novel composite bioinks made by blending adipose decellularized extracellular matrix with plasma and human dermal fibroblasts. Rheological and microstructural assessments of the composite hydrogels supported post-printing cell viability and proliferation over time. Embedded fibroblasts expressed steady concentrations of extracellular matrix proteins, including type 1, 3 and 4 collagens and fibronectin. ELISA assessments, multiplex protein arrays and ensuing bioinformatic analyses revealed paracrine activities corresponding to wound-healing activation through the modulation of inflammation and angiogenesis. The two modalities of advanced dressings, differing in platelet number, showed differences in the release of inflammatory and angiogenic cytokines, including interleukin 8 (IL-8), monocyte chemotactic protein 1 (MCP-1), vascular endothelial growth factor (VEGF) and hepatocyte growth factor (HGF). The conditioned media stimulated human-dermal-cell proliferation over time. Our findings open the door to engineering the microenvironment as a strategy to enhance healing.

## 1. Introduction

Skin is the first line of defense against many types of infections and diseases. Acute skin injuries, in healthy people, repair through a sequence of complex, constitutively active signaling pathways. However, in patients with comorbidities, a large number of wounds become chronic as they fail to repair within three months [1]. Diabetes, venous stasis, radiation or paralyses are common risk factors for complex wounds. In these patients, healing mechanisms fail to progress through the different stages and ulcers become stagnant. Wound chronicity, referred to as silent epidemic, is an overwhelming economic and medical burden for healthcare systems [2,3]. Common biological features of chronicity include uncontrolled inflammation and loss of the dermal cells’ ability to respond to reparative stimuli [4].

The current standard of care (SOC) addresses several specific aspects related to wound etiology and involves the control of moisture in the wound bed through the careful personalized selection of dressings [5], which, theoretically, provide the conditions for proper cell/protein interactions committed to healing. However, closure rates in complex wounds are low [6]. On the other hand, cell-based wound dressings are commercialized and investigated to replace the current SOC [7]. Most cell-based dressings utilize a hydrogel scaffold upon which cells are seeded. There are three general categories, which include (1) amniotic and placental membranes (e.g., dehydrated (EpiFix^®^, Marietta, GA, USA), or cryopreserved (Grafix*, Osiris, Columbia, MD, USA)) [6]; (2) human allogeneic skin cells seeded in bovine type 1 collagen (e.g., OrCel^®^ (Ortec International NY, USA), Apligraf^®^, Organogenesis Inc., Canton MA, USA) [8] or polyglactin mesh scaffold (Dermagraft^®^, Organogenesis Inc., Canton MA, USA) [9]; and (3) allogeneic fibrin patch with platelets and leukocytes (LeucoPatch^®^ (Reapplix, Birkerød Denmark)) [10]. Their efficacy is better than the SOC, but there are still opportunities for refinement, as efficacies range between 31 and 50% of closure [7]. In addition to the limited efficacy, higher upfront costs limit their integration in clinical practice [7]. The latter can be overcome by the adoption of novel manufacturing technologies, such as 3D bioprinting [11]. In particular, extrusion bioprinting is compatible with almost all hydrogels and can be easily scaled up to print at a reasonable cost [12,13,14]. 

Upgrading dressing manufacturing can help to meet the urgent clinical need created by ulcer chronicity, helping to mitigate obstacles to healing and, in doing so, restoring the skin barrier function in complex situations. Moreover, bioink customization can be achieved by the careful selection of autologous or homologous components following a biomimetic approach.

Even so, advanced dressings can fail to boost healing mechanisms because they encounter a hostile microenvironment rich in inflammatory cytokines and enzymes, such as collagenase, gelatinases, stromelysins and cell-membrane-associated MMPs (Matrix Metalloproteinases), which contribute to the rapid clearance of growth factors and cytokines in the wound area and are detrimental for cell viability. Therefore, engineering the microenvironment has been regarded as a paramount trend to enhance cell survival and burst-healing mechanisms by the constant release and diffusion of signaling proteins [15].

Several novel bioinks for skin engineering have been developed in the past two years [16]. However, existing bioinks cannot meet the complex needs of difficult-to-heal wounds [11]. An ideal scenario would present the cells within advanced dressings in a natural microenvironment that exhibits similar characteristics to a healing tissue. In this work, biomimetic hydrogels formulated as smart bioinks for manufacturing new modalities of advanced dressings were created by blending porcine decellularized adipose matrix (pDAM2) [17] and alginate with platelets and plasma growth factors at two concentrations and human dermal cells. The proteomic analysis of pDAM2 confirmed that several proteins, specifically, collagens, proteoglycans, glycoproteins and affiliated proteins, were preserved. The complete proteomic analysis of pDAM2 material has been previously reported by the team [17]. Besides, pDAM2 hydrogels can provide mechanical support to enhance cell attachment and modulate cell behavior, regulating cell phenotype and function, as well as infection. Actually, since the Egyptians [18], bandages made with grease from animal fat have been used as a barrier to bacteria to treat battle injuries [19]. On the other hand, by adding PRP (platelet-rich plasma) to bioink (dressing), we provided active biological molecules to drive cell activities towards the enrichment of healing mechanisms. Finally, by blending pDAM2, plasma and human dermal fibroblasts, our advanced dressings could mimic cell-to-matrix and cell-to-environment interactions, paramount for physiologically relevant cell functions. 

Within the clinical context, according to a recent meta-analysis, PRP-assisted fat grafting is effective in soft-tissue augmentation [20]. In particular, diabetic foot ulcers treated with fat and PRP showed increased vessel density and graft survival [21]. Although fat combined with PPP has been poorly investigated in humans, research in nude mice revealed that both PRP and PPP enhanced fat-graft survival, although PRP had stronger angiogenic effects [22].

The two main goals of this research study are: (1) to formulate and compare two different modalities of composite bioink made with natural extracellular matrix (ECM)-derived hydrogels and laden with human dermal fibroblasts, valid for extrusion bioprinting; and (2) to examine the biological and functional properties of bioprinted constructs and their capabilities to create an optimal molecular microenvironment over time for wound healing.

## 2. Results

### 2.1. Rheological and Microstructural Properties of the Composite Bioinks

For extrusion-based 3D printing, the material must have adequate zero-shear viscosity and shear-thinning properties to extrude smoothly through a narrow nozzle. The pDAM2: PRP/ALG and pDAM2: PPP/ALG bioinks showed shear-thinning behavior in a shear-stress range of 1–1000 s^−1^—viscosity is a decreasing function of the shear rate. Viscosity values at shear rates of 1 s^−1^ for the pDAM2: PRP/ALG and pDAM2: PPP/ALG hybrid bioinks were measured at 6.30 and 8.27 Pas, respectively. The viscosity of the hybrid bioinks showed medium values compared with raw bioinks DAM2, PRP/ALG and PPP/ALG (Figure 1A).

Because good fluidity during bioink extrusion and gelling ability just after deposition are essential for bioprinting, the rheological behavior of the hydrogels was investigated. When the rheological properties were measured under oscillating conditions after gelification, the hybrid hydrogels exhibited gel-like properties, with the storage modulus (G′) being higher than the loss modulus (G″) (Figure 1B). Thus, after gelation, the hydrogels retained their shape and form, which is a prerequisite for the fabrication of 3D-bioprinted dressings. The pDAM2 hydrogel showed the lowest storage modulus; moreover, the PRP/ALG and PPP/ALG hydrogels exhibited a similar rheological profile in terms of storage and loss modulus and revealed larger storage moduli than the hybrid hydrogels (pDAM2:PRP/ALG, pDAM2:PPP/ALG).

Scanning electron microscopy of the pDAM2 hydrogels showed a randomly oriented fibrillar structure, with an average fiber diameter of less than 100 nm and interconnecting pores. The PRP/ALG and PPP/ALG hydrogels exhibited microstructures with many irregular aggregates. However, the images of the pDAM2: PRP/ALG and pDAM2: PPP/ALG hybrid hydrogels revealed that those aggregates were covered by the fibrillar structure of the pDAM2 hydrogels (Figure 2).

Regarding the advantages in bioprinting processes, the behavior of the hydrogel blends (pDAM2: PRP/ALG and pDAM2: PPP/ALG) was individually superior to the plasma bioinks and allowed us to reduce the diameter of the extrusion needle (from 20 G to 22 G), improving filament homogeneity, without detrimental consequences in terms of cell viability.

Before bioprinting, cytocompatibility was assessed. The HDFs encapsulated within these constructs were metabolically active in the formulation of both hydrogels. Both blends, pDAM2: PRP/ALG and pDAM2: PPP/ALG, stimulated higher metabolic activity than the single pDAM2 hydrogel (Appendix A). 

### 2.2. Three-Dimensional Bioprinting Procedure, Post-Printing Cell Viability and Proliferation in Wound Dressings

Following the methodology described before [23], proper 3D-printing resolution and dispensing uniformity were achieved with both pDAM2:PRP/ALG and pDAM2:PPP/ALG bioinks, obtaining advanced wound dressings with high reproducibility. This was due to filament uniformity during extrusion and maintenance of the construct stability with 100% infill. Further structural integrity of the dressings was achieved by physical cross-linking using CaCl2 (Figure 3A). The results showed a high number of HDF cells viable after bioprinting with pDAM2: PRP/ALG and pDAM2: PPP/ALG (Figure 3B,C), whereas a very low number of dead cells was observed in all tested dressings. Staining for cell viability and cell density revealed that both bioink modalities had comparable cytocompatibility and bioactivity to promote cell growth. Changes in cell density and morphology over time confirmed that the bioinks allowed cells to adhere and grow in an ideal 3D microenvironment.

### 2.3. Expression of Extracellular Matrix Proteins by Embedded Dermal Fibroblasts

The HDFs embedded within the advanced dressings showed high expression of FN and moderate expression of COL1A1, COL1A2 and COL3A1 relative to GAPDH. On the other hand, the expression of COL4A1 and COL4A2 was low and ELN expression was under the detection limits. Repeated measurements of gene expression showed steady levels of expression, without statistically significant changes over time in any of the analyzed molecules (Figure 4).

### 2.4. Paracrine Potential of Wound Dressings: Signaling Protein Expression over Time

To evaluate the effects of manufactured dressings on the wound microenvironment, we investigated the release of healing signaling molecules over time. Fibroblasts embedded in either pDAM2: PRP/ALG or pDAM2: PPP/ALG synthesized and released wound-healing cytokines over 11 days; protein concentrations were normalized with protein data at 1 h. In doing so, we obtained folds relative to one-hour cultures (shown in Appendix A, respectively). Using core IPAs, we found that the “wound healing” canonical pathway was significantly enriched in our data. Figure 5A depicts important molecules in this pathway. Z-score values predicted that wound healing was activated over time by the molecules released from the construct (Figure 5B). The Z-score increased over time for the pDAM2: PRP/ALG dressings (4 d, Z = 0.365; 7 d, Z = 1.616; 11 d, Z = 3.413), while the maximum Z-score for pDAM2: PPP/ALG was achieved at 7 d (4 d, Z = 1.061; 7 d, Z = 2.611; 11 d, Z = 1.257). In parallel, the Z-score for paracrine activation of cell signaling and viability displayed a similar pattern of variation, with differences between both dressings (Figure 5C).

To confirm the predictions obtained by the IPA algorithms, we used ELISAs to perform individual assessments of relevant cytokines involved in cell signaling and modulation of inflammation and angiogenesis. IL-8 is a pleiotropic interleukin involved in cell activation and movement; both modalities of advanced dressings showed significant synthesis and release of IL-8 over time (*p* < 0.001). Maximum release was achieved after 4 days without additional synthesis from day 4 to day 11 in any of the bioink variants (*p* ≤ 0.001) (Figure 6A). On the other hand, MCP-1, also known as CCL-2, was not detected in acellular scaffolds at any time point (Figure 6B). However, the HDFs embedded within the pDAM2:PRP/ALG dressings synthesized increasing concentrations of MCP-1 over time (*p* < 0.001); maximum concentrations were achieved after 11 days of culture, representing a 70-fold increase compared with 1-hour concentrations (26.72 ± 1.14 ng/mL at 11 days and 0.372 ± 0.008 ng/mL at 1 hour). Instead, the pDAM2: PPP/ALG dressings released MCP-1 after four days but did not show additional synthesis over time. The amount of MCP1 released by the pDAM2: PRP/ALG dressings was significantly higher at 11 days (26.716 ± 1.14 ng/mL versus 8.92 ± 0.52 ng/mL; *p* < 0.001).

We quantified VEGF and HGF as an index of angiogenesis. There was a significant synthesis of VEGF and HGF by both advanced dressings, i.e., manufactured with either the pDAM2: PRP/ALG or pDAM2: PPP/ALG bioink, with *p* < 0.001 for both dressings and both proteins. Of note, cells within the pDAM2: PPP/ALG dressings released higher concentrations of VEGF and HGF after 7 and 11 days (Figure 6C,D).

On the other hand, we measured RANTES (CCL5) and PDGF, which are abundant platelet proteins, here provided by the PRP component of the prepared bioink. The pDAM2: PPP/ALG constructs did not show any synthesis/secretion of RANTES and PDGF-BB. Remarkably, RANTES and PDGF-BB released from the PRP dressings without cells showed higher concentrations than those released by the constructs with cells, pointing out protein consumption by the embedded fibroblasts. 

### 2.5. Indirect Co-Cultures: Dermal-Cell Proliferation

The conditioned media harvested over time from the two modalities of advanced dressings (differing in platelet number) stimulated human-dermal-cell proliferation, as revealed by indirect co-cultures. The media harvested from the bioprinted constructs manufactured without cells was used as control. The conditioned media from cells within the pDAM2: PRP/ALG dressings had a more-potent proliferating activity than the conditioned media from the pDAM2: PPP/ALG dressings (*p* = 0.001) (Figure 7). Although the proliferation rate was optimal for both dressing modalities, those manufactured with pDAM2: PRP/ALG reached higher proliferation rates with the conditioned media harvested after 7–11 days of in vitro maturation.

## 3. Discussion

In this study, we propose, for the first time, the combination of DAM, plasma and dermal fibroblasts to create bioink for advanced-dressing manufacturing. We formulated two modalities of smart bioink, which meet the requirements for extrusion bioprinting, by blending pDAM2 with two different formulations of plasma ink (PPP/ALG and PRP/ALG). The latter mimic the pool of soluble factors present in the wound, as previously reported [23], thereby adding further functionalities to the pDAM2 bioink component and favoring fibroblast functions. As shown, both hydrogel composites showed good cell compatibility with dermal fibroblasts and supported their biological paracrine functions. Using multiplex protein analyses, followed by data analyses by means of cloud-based software (QIAGEN IPA^®^, Redwood City, CA; USA), we found that dermal cells embedded within the bioprinted dressings synthesized and released, over time, a large pool of signaling proteins involved in the canonical wound-healing pathway (Figure 1). Moreover, the secreted molecular pool favored cell-to-cell signaling and interactions, suggesting that cell activities within the bioprinted dressings can drive regenerative mechanisms through paracrine signaling. We believe it is reasonable to hypothesize that signaling cytokines synthesized and released to the extracellular milieu contribute to creating a dynamic optimal microenvironment for healing.

In a recent systematic review, we used technology readiness levels (TRLs), as a valid metric to assess the evolution and readiness of extrusion bioprinting for skin applications [24]. We evidenced the immaturity of bioink developments (TRL 3–4) in contrast to the rapid advancements of bioprinter devices (TRL 9) [25]. However, in the last two years, there has been an expansion of bioink research for wound healing and novel blends have been proposed [16,26,27,28]. Adding to bioink advancements for extrusion bioprinting, we blended two natural hydrogels, sharing a common crosslinking mechanism for the post-printing stability of the construct. We used, firstly, decellularized extracellular matrix from porcine adipose tissue and, secondly, a previously developed plasma bioink with two different platelet concentrations [23]. Adipose tissue has high potential as an abundant source of ECM to prepare biomaterials designed for the growth, differentiation and preservation of different cell phenotypes [29,30]. The techniques commonly used to remove cellular components from the extracellular matrix include physical, chemical and biological agents [30,31]. In this study, we decellularized porcine adipose tissue using our chemical protocol [17]. 

The separate use of either DAM or plasma as bioink components is not new; however, our study reports, for the first time, the behavior of composite bioink. By blending DAM with plasma and platelet signaling proteins, we reinforced the functionalities of the embedded cells in the bioprinted constructs. Moreover, our model resembles the microenvironment of a healing wound. As shown in this study, adipose ECM provides a porous support for cell adhesion and allows nutrients to circulate, permitting long-term cell culture in vitro, while the richness in molecular diversity of plasma bioink involves participation in biological processes, including inflammation, angiogenesis, cell proliferation and synthesis/maintenance of the ECM [32]. 

Previous research has examined the functionalization of wound dressings with molecules targeted to specific processes. In particular, membranous ECM-based scaffolds, such as acellular dermal matrix or small intestinal submucosa, were loaded with antibacterial nanoparticles, antiseptic peptides, or anti-inflammatory agents or growth factors (GFs) [33]. For example, bFGF (basic fibroblast growth factor) or VEGF was added to some dressings in order to reinforce granulation-tissue formation in the initial healing phase [34]; specifically, porcine decellularized adipose tissue loaded with VEGF showed enhanced angiogenesis in vivo and in vitro [35]. Likewise, the incorporation of rhEGF (recombinant human epidermal growth factor) or rhPDGF was used to stimulate cell proliferation and growth in a subsequent healing phase [36,37]. Alternatively, membranous ECM dressings loaded with anti-inflammatory molecules, such as TNF-a siRNA and miRNA146am, could drive macrophage polarization towards healing [38]. Instead, our efforts focused on the incorporation of cells within the dressings, thereby allowing functional proteins to be continuously synthesized. 

The best cell phenotype for complex wounds is unknown [39]. We used dermal fibroblasts because of their safety [40] and their extensive contribution to wound healing. Nevertheless, the participation of local dermal fibroblasts in the healing mechanisms of complex wounds is generally hindered, because they present premature stress-induced senescence, decreased proliferative ability and impaired ability to respond to GFs and cytokines [4,41,42,43]. Based on this premise, the clinical application of exogenous dermal fibroblasts in complex wounds, such as DFU (Diabetic foot ulcer), showed good healing rates compared with foam dressings [44,45]. Moreover, among the advanced dressings available on the market, the human-fibroblast-derived dermal substitute known as Dermagraft^®^ (Organogenesis Inc., Canton MA, USA) (it includes dermal fibroblasts, within a bioabsorbable polyglactin mesh) showed 44% and 71.4% healing in DFU after 8 applications and 50% of wound closure in vascular leg ulcers after 4 applications [46,47]. 

In recent years, two different plasma formulations have been used to functionalize biomaterials for tissue engineering [23,48,49,50], specifically, PRP and PPP, equivalent to freeze-frozen plasma (FFP) in transfusion medicine. Cubo et al. pioneered the use of PPP as a bioink component for skin repair. Because of the easier production and yield of larger volumes of PPP than PRP from the same blood volume, PPP is broadly used when plasma-based bioinks are created to bioprint tissue equivalents for disease models and for tissue additive biofabrication [51]. However, molecular differences among plasma products can have enormous implications in both clinical applications and tissue engineering. In physiology, platelets and their secretome, mainly GFs and other cytokines released upon activation, play an important role in healing processes. In fact, platelets are present from the initial hemostatic phase in tissue healing, a fact that inspired the therapeutic application of platelets decades ago [52].

In a recent study [53], we showed that the molecular intricacy of PRP by far exceeds that of PPP. In fact, platelet secretome contains a pool of chemokines, growth factors and interleukins, which are synthesized in the bone marrow by their parent cell, the megakaryocyte. Our results are consistent with this information. While the fibroblasts embedded in both bioinks showed similar expression of structural proteins (collagens and fibronectin), we found differences in the synthesis and release of crucial cytokines over time, i.e., MCP-1, RANTES and IL-8, and angiogenic GFs, such as VEGF and HGF. Interestingly, diabetic-foot skin loses the ability to produce chemokines; the application of MCP-1 in diabetic wounds promoted healing in a murine model [54].

However, delayed healing is not attributed to a single cytokine but to alterations in molecular networks. To approach this complexity and given the importance of paracrine signaling, we examined the composition of media conditioned by advanced dressings kept under culture conditions over time. By examining the differential protein expression over 11 days and 1 h, we observed the dynamic synthesis and release of GFs, chemokines and cytokines, which were able to modify the wound microenvironment. In addition, we found that both dressings could achieve a desired biological effect, i.e., local cell proliferation to fill the wound discontinuity. In fact, the conditioned media harvested at different time points stimulated dermal-cell proliferation. Essentially, the interactive dressings showed a promising basis for microenvironment engineering through the paracrine actions of the cells embedded in the bioprinted dressings.

This study is mainly focused on investigating the paracrine response of human dermal fibroblasts to differences in bioink composition and our findings open the door to the engineering of the wound microenvironment as a strategy to enhance healing. This work uncovers inflammatory modulation (e.g., IL-8, MCP-1) and potential promotion of angiogenesis (e.g., VEGF, HGF) through the dynamic synthesis of cytokines and growth factors. However, further in vivo experiments analyzing their behavior in two particular environments commonly found in the clinical scenario (inflammation and ischemia) are needed to establish precise indications and to validate the proposed composite bioinks. Moreover, analyses of in vivo biodegradability and the rate of wound healing are not addressed here; these are major limitations that do not allow us to determine the true value of our bioprinted constructs in wound repair. Our future studies will utilize these combinational bioinks to investigate the in situ delivery of advanced dressings with handheld devices [34,55].

## 4. Materials and Methods

### 4.1. Individual Components of the Composite Bioinks 

#### 4.1.1. Preparation of Porcine Decellularized Adipose Matrix (pDAM2) 

Porcine decellularized adipose matrix (pDAM2) was prepared following a methodology previously published by the group [17]. Briefly, clean adipose tissue, supplied by a local food company (JAUCHA S.L., Navarra, Spain) was creamed using a beater and stored at −20 °C. For tissue decellularization, creamed tissue was defrosted, homogenized with ultrapure water (Polytron PT3100, Yakarta, Indonesia) at 12,000 rpm for 5 min and centrifuged (5000 rpm for 5 min) for protein precipitation. After lipid removal, the protein pellets were treated with isopropanol, 1% (*v*/*v*) triton x-100 and 0.1% (*v*/*v*) ammonium hydroxide (Merck Life Science SL, Darmstadt, Germany). Pellets were washed with phosphate buffer saline (PBS; Merck Life Science SL, Darmstadt, Germany) supplemented with 1% (*v*/*v*) antibiotic antimycotic solution (Gibco-BRL, Paisley, UK) followed by a final wash with Ultrapure milli Q water. The pDAM2 material was lyophilized, milled (with a mixer mill; Retsch MM400; Haan, Germany) and stored at 4 °C in a vacuum desiccator. The effective decellularization was confirmed by a remnant-DNA analysis [17].

To make pDAM2 printable, the material was enzymatically digested at a concentration of 15 mg/mL in a solution of 1 mg/mL porcine pepsin (Sigma Aldrich, St. Louis, MO, USA) in 0.1 N HCl under a constant stir rate for 48 h at room temperature (RT). The pH and salt concentration of the digested pDAM2 were adjusted to 7.2–7.4 and kept on ice. 

#### 4.1.2. Preparation of Plasma Inks

Venous blood was obtained from local blood-bank donors (*n* = 6) after obtaining informed consent (CES-BIOEF 201907). Two different plasma formulations, platelet-poor plasma (PPP) with a platelet count 10-fold below peripheral blood and platelet-rich plasma (PRP) with a platelet concentration 4-fold above peripheral blood, were produced from each donor and pooled. To create PPP/ALG and PRP/ALG inks, freeze-thawed PPP and PRP were blended with alginic acid sodium salt (#180947; Sigma–Aldrich, St. Louis, MO, USA) prepared in pre-warmed DMEM (previously autoclaved solution 8%, *w*/*v*). 

### 4.2. Elaboration and Characterization of the Composite Bioinks

The plasma inks were pre-crosslinked with 20 mM CaCl_2_ (prepared in DMEM) and kept at 37 °C overnight prior to mixing with pDAM2 in a 3:1 ratio. To ensure the absence of any cytotoxic effect of the hydrogels, the blends were manually loaded with primary human dermal fibroblasts (HDFs; 2 × 10^6^ cell/mL) (Innovative Technologies in Biological Systems, Innoprot, Derio, Spain) and cultured for seven days. Cytotoxicity was examined using a WST-1 Colorimetric Assay (Cell Proliferation Reagent; Roche 5015944001; Merck Life Science SL, Madrid, Spain) at 1, 5 and 7 days of culture.

#### 4.2.1. Rheological Properties 

The viscoelastic properties of the composite bioinks were examined on a parallel-plate geometry (200 mm diameter steel with a gap of 1 mm) using the Thermo Scientific HAAKE RheoStress 6000 Rotational Rheometer (Thermo Scientific, Karlsruhe, Germany). To assess the viscosity, steady-shear sweep analyses of the pDAM2: PRP/ALG and pDAM2: PPP/ALG combinations of composite bioinks were performed at a constant temperature of 22 °C. A dynamic-frequency sweep analysis was conducted in duplicate to measure the frequency-dependent storage (G′) and loss (G″) moduli of the pDAM2: PRP/ALG and pDAM2: PPP/ALG hydrogels in the range of 1–1000 rad s–1 at 2% strain after the incubation step. Individual bioink components, pDAM2 and PRP/ALG or PPP/ALG, were used as control material.

#### 4.2.2. Scanning Electron Microscopy (SEM) 

The ultrastructure of the composite ink (pDAM2 blended with PRP/ALG) was examined in parallel with the individual components (pDAM2 and PRP/ALG) by SEM. Samples were dehydrated using a series of ethanol gradients as previously described [56]. After the gold spattering of hydrogels, images were captured using ultrahigh-resolution SEM (JEOL JSM-5910 LV; Tokyo, Japan).

### 4.3. Three-Dimensional Bioprinting of Two Advanced-Dressing Modalities 

Dressings were manufactured using a desktop syringe-based extrusion bioprinter (DOMOTEK BIO, Donostia, Spain). Five-milliliter syringes loaded with the composite bioinks were incubated at 37 °C for 10 min before printing. Regarding the printability of the bioinks, this was previously evaluated independently for each hydrogel-based ink, as well as for the blend, following the protocol described by Del Amo et al. (2021) [23]. The optimization of the extrusion parameters included the nozzle selection and the definition of the layer height (0.4 mm), as well as the total number of layers (2) in the constructs. Afterwards, the bioinks were extruded at 10 mm/s through a 0.41 mm blunt needle at 37 °C. The bioprinted constructs, with disk-shape morphology (8 mm diameter and 0.4 mm thickness), were deposited on 24-well culture plates. Post-printing gelation was achieved by crosslinking with 100 mM CaCl_2_ for 20 min at 37 °C.

### 4.4. Cell Viability within the Advanced Dressings

Live–dead (Calcein-AM and Propidium iodide; Invitrogen™, Waltham, MA, USA) cell staining was used to assess cell viability at 0, 4, 7 and 11 days of culture. The bioprinted cell-laden constructs were washed in PBS 1x (Gibco™, Waltham, MA, USA) and stained with Calcein-AM (5 µM in PBS 1x; Invitrogen™, Waltham, MA, USA) for 30 min at 37 °C and 5% CO_2_ and Propidium iodide staining (1 mg/mL; Sigma-Aldrich, Burlington, MA, USA) for 5 min at 37 °C and 5% CO_2._ Viable (colored in green) and non-viable cells (colored in red) were observed by fluorescent microscopy (BX51 with DP74 camera and Stream Basic Software; Olympus, Tokio, Japan).

### 4.5. Cell Proliferation over Time 

Proliferation was assessed by DNA quantification. At each time interval, the constructs were washed in PBS 1x (Gibco™, Waltham, MA, USA) and stored frozen at −80 °C until analysis. For DNA quantification, the constructs were digested in 20 UI/mL Papain solution (Worthington Biochemical Corporation, Lakewood, NJ, USA) at 65 °C for 1 h and total DNA (ng DNA/mg construct) was determined using a Quant-iT Pico Green^®^ dsDNA kit (P7581; Invitrogen^TM^; Waltham, MA, USA), following the manufacturer’s instructions.

### 4.6. Gene-Expression Analysis by RT-ddPCR

RNA isolation from the cells embedded within the two modalities of advanced dressings was performed by Trizol treatment and its purity was assessed by the OD ratio (OD_260_/OD_280_) using a NanoDrop 2000 (Thermo Scientific™, Waltham, MA, USA). The expression of dermal extracellular matrix proteins was assessed over time using one-step RT-ddPCR (ddPCR stands for droplet digital PCR). The reaction mix was prepared in a final 21 μL volume by mixing 5 μL of ddPCR™ Supermix One-Step (Bio-Rad Laboratories, Hercules, CA, USA), 20 U/μL reverse transcriptase, 15 mM dithiothreitol (DTT), 900 nM target primers and 250 nM probes, and 1.25–5 ng RNA. A volume of 20 μL of the reaction mix was loaded into a DG8 Cartridge for automated droplet generation (QX200 Droplet Generator; Bio-Rad Laboratories, Hercules, CA, USA). The droplets were then transferred into a 96-well plate and heat-sealed before amplification using an iQX One Droplet Digital PCR System (Bio-Rad Laboratories, Hercules, CA, USA). The PCR thermal cycling conditions were: hold at 25 °C for 3 min, reverse transcription at 50 °C for 60 min, denaturation at 95 °C for 30 sec, annealing at 55.5–65 °C for 2 min and then 98 °C for 10 min for enzyme deactivation. A QX200 Droplet reader and QuantaSoft™ Software (Bio-Rad Laboratories, Hercules, CA, USA) were used in order to read droplets and for data processing. 

Appendix A shows the primer sequences used for the gene-expression analyses. *GAPDH* (glyceraldehyde 3- phosphate dehydrogenase) was used as housekeeping gene and the studied genes were *COL1A1* (collagen type I alpha 1 chain), *COL1A2* (collagen type I alpha 2 chain)*, COL3A1* (collagen type III alpha 1 chain), *COL4A1* (collagen type IV alpha 1 chain)*, COL4A2* (collagen type IV alpha 2 chain)*, ELN* (elastin) and *FN* (fibronectin). Appendix A shows the conditions for RT-PCR.

### 4.7. Molecular Characterization of the Conditioned Media, Ingenuity Pathway Analysis (IPA) and ELISA

Conditioned culture media were harvested over time (up to 11 days) to examine the paracrine properties of the constructs by means of multiplex protein arrays, which detect human inflammatory cytokines and growth factors (126QAH-CAA-1000-1 Human Cytokine Array Q1000 Quantibody Human Cytokine Array 1000 Kit; RayBiotech, Peachtree Corners, GA, USA). The arrays were performed according to the manufacturer’s instructions and the results were quantified against array positive controls. Array scanning was performed by the manufacturer’s service and data were analyzed using Quantibody Q-Analyzer Software version 8.40.4 (RayBiotech, Peachtree Corners, GA, USA).

The proteins studied in the arrays were annotated and classified according to their GO functions. In order to analyze dermal fibroblast secretome, fold changes were calculated using 1 h secretome as reference. We evaluated the data set obtained from the multiplexing platform in the context of a large, structured collection of observations in various experimental settings with nearly 5 million findings either manually curated from the biomedical literature or integrated from third-party databases using Ingenuity Pathway Analysis Software (IPA; Qiagen, Redwood City, CA, USA). Signal pathway networks and canonical pathways were predicted using IPA algorithms. 

Figure 8 depicts the main functional features of the advanced dressings that we examined in this research study. (1) Encapsulated cells proliferated within the 3D-bioprinted constructs (evaluated by DNA quantification); moreover, using indirect co-cultures, we measured their paracrine activities on human-fibroblast proliferation. (2) The expression of dermal extracellular matrix components (including various collagen types, fibronectin and elastin) by the encapsulated cells was assessed over time by digital RT-PCR. (3) In parallel, multiplexing analyses of signaling proteins, synthesized by the encapsulated cells and released to the conditioned media, were performed to understand the paracrine activities and the molecular pattern of the wound microenvironment.

ELISA assessments: By using ELISAs, we studied the levels of human RANTES (CCL5) (900-K33; Peprotech Inc, Rocky Hill, NJ, USA), human MCP-1 (CCL2) (900-K31; Peprotech Inc., Rocky Hill, NJ, USA), human IL-8, human PDGF-BB (900-K04; Peprotech Inc., Rocky Hill, NJ, USA), human HGF (ELH-HGF; RayBiotech, Peachtree Corners, GA, USA) and human VEGF (900-K10; Peprotech Inc., Rocky Hill, NJ, USA). To confirm the de novo synthesis of selected proteins by the embedded dermal fibroblasts, acellular scaffolds were cultured in parallel; we quantified the signaling proteins released from acellular scaffolds over time and subtracted them from the proteins released from cellular scaffolds.

### 4.8. Activity Assays, Indirect Co-Cultures

The proliferation of dermal fibroblasts incubated in the conditioned media (CM) collected from both dressing modalities was evaluated at 1 h and 4, 7 and 11 days. After collection, cell media were centrifuged for 8 min at 9600× *g* at room temperature and then filtered through spin centrifuge filters (0.45 µm in pore size). Each conditioned medium was diluted at 1:1 with EGM-2 (endothelial cell growth medium) (Lonza, Switzerland). Before the experiment, the cells were starved in serum-free medium for 24 h. Then, 1 × 10^5^ and 3 × 10^4^ cells·mL^−1^ were seeded into 96-well plates and the plates were incubated for 24 h and 7 days, respectively. XTT solution (Sigma–Aldrich, Burlington, MA, USA) was added to the cells and the incubation was continued for 4 h. The absorbance of each well was measured at a wavelength of 490 nm and 5% PRP or 5% PPP supernatants were used as cell culture supplements for the normalization of proliferation data.

### 4.9. Statistical Analysis

Data are expressed as the mean ± SD, unless otherwise specified. A repeated-measure ANOVA was performed for multiple time points. Non-normal data were examined using the Friedman test, followed by Wilcoxon paired comparisons. For all comparisons, statistical significance was set at *p* ≤ 0.05. Data were analyzed using SPSS for Windows version 18.0 (SPSS, Inc., Chicago, IL, USA).

## 5. Conclusions

Our study reports the formulation of novel bioinks concurrently with the practicability of 3D bioprinting for advanced-dressing manufacturing. The formulation of natural composite hydrogels created by combining adipose extracellular matrix and two different blood derivatives loaded with dermal fibroblasts is amenable to advanced wound-dressing manufacturing through extrusion bioprinting with excellent cell viability. In addition, we reveal the dynamic up-regulation of a large pool of healing proteins with promising corrective roles in stagnant wounds, as predicted by bioinformatic analyses.

## Figures and Tables

**Figure 1 ijms-23-02836-f001:**
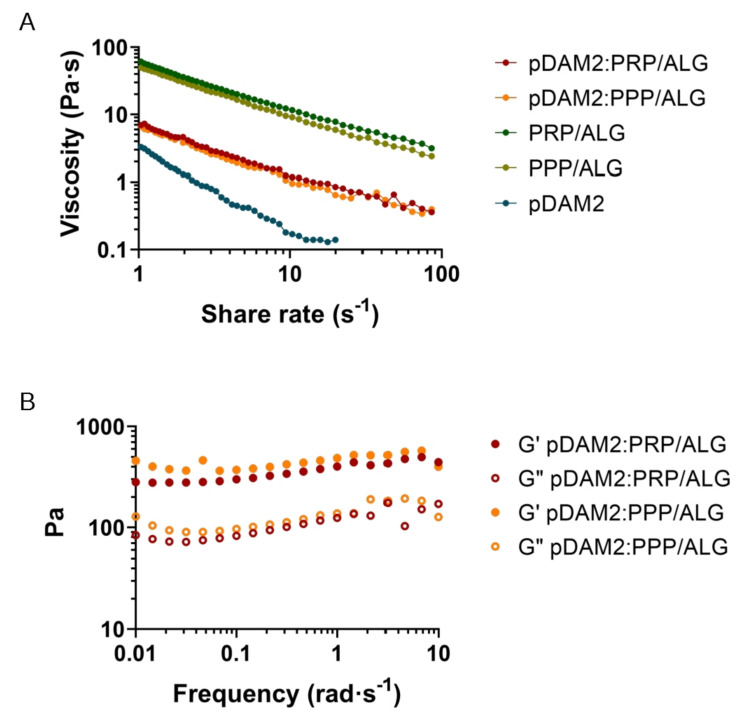
Rheological behavior: (**A**) flow curves of pDAM2: PRP/ALG and pDAM2: PPP/ALG inks and the individual components; (**B**) dynamic moduli at varying frequency at 37 °C of hydrogels (storage and loss moduli measured by frequency sweep tests of pDAM2: PRP/ALG and pDAM2: PPP/ALG inks).

**Figure 2 ijms-23-02836-f002:**
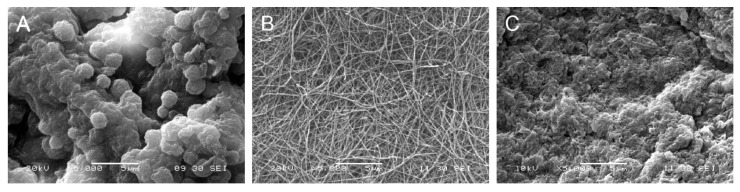
Scanning electron microscopy (SEM) images of: (**A**) PRP/ALG, (**B**) pDAM2 and (**C**) pDAM2: PRP/ALG.

**Figure 3 ijms-23-02836-f003:**
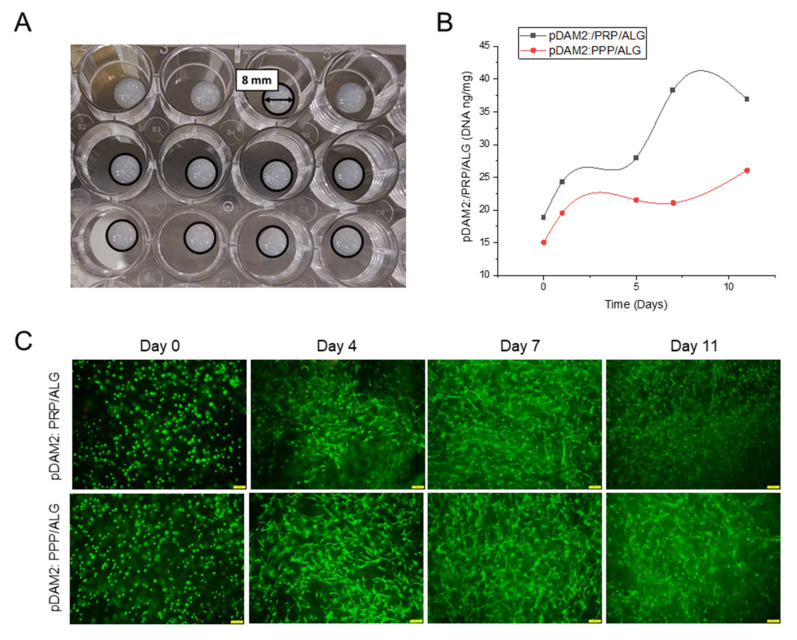
(**A**) Macroscopic image of cell-laden constructs bioprinted on 24-well plates. (**B**) HDF proliferation (DNA ng/mg construct) within a period of 11 days of culture. (**C**) HDF viability at 0, 4, 7 and 11 days of culture after bioprinting with pDAM2:PRP/ALG and pDAM2:PPP/ALG. Green corresponds to calcein-AM staining of life cells and red corresponds to Propidium iodide staining of dead cells (scale bar = 100 µm).

**Figure 4 ijms-23-02836-f004:**
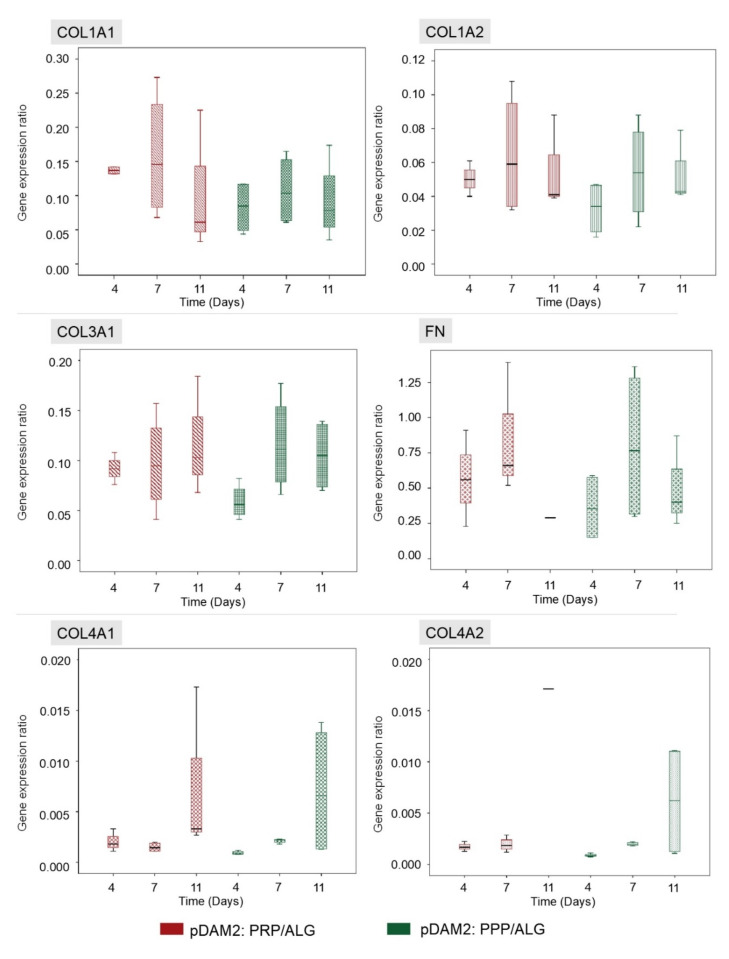
HDFs embedded within both modalities of advanced dressings showed moderate expression of COL1A1, COL1A2 and COL3A1 relative to GAPDH, without variations over time. The expression of fibronectin was high in both dressing types (RNA from the 11-day construct was lost), whereas the expression of COL4A1 and COL4A2 was low. There were no significant changes in gene expression over time and no differences between the two dressing modalities.

**Figure 5 ijms-23-02836-f005:**
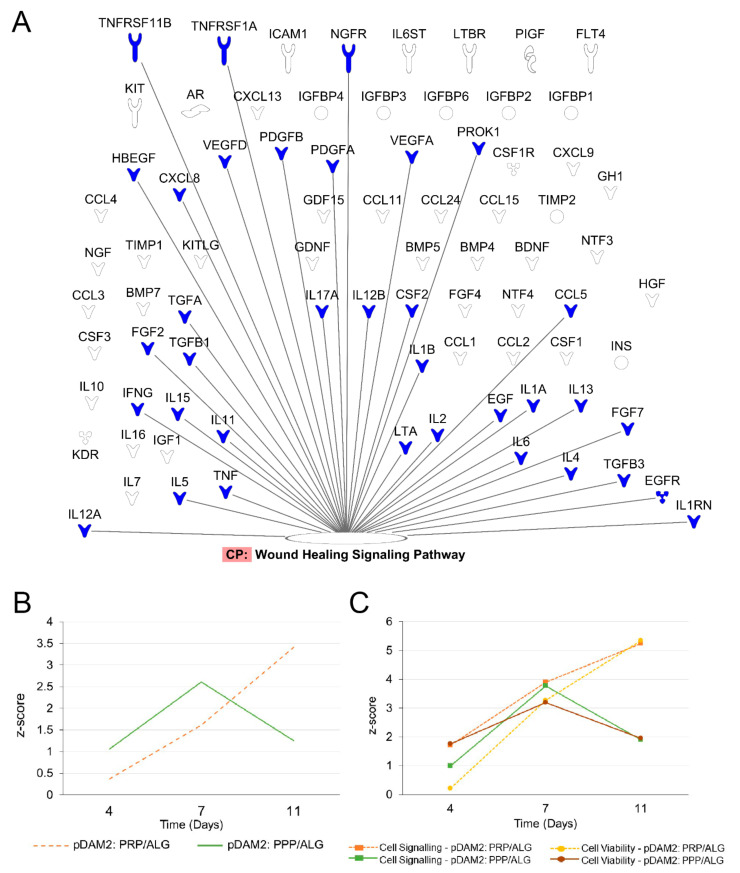
Signaling pathways enriched in the data obtained from the analysis of the conditioned media. (**A**) Signaling molecules synthesized by encapsulated cells over time that were involved in the wound-healing signaling pathway. (**B**,**C**) pattern of Z-score outcome over time predicting activation of the wound-healing signaling pathway (**B**), cell signaling and cell viability (**C**).

**Figure 6 ijms-23-02836-f006:**
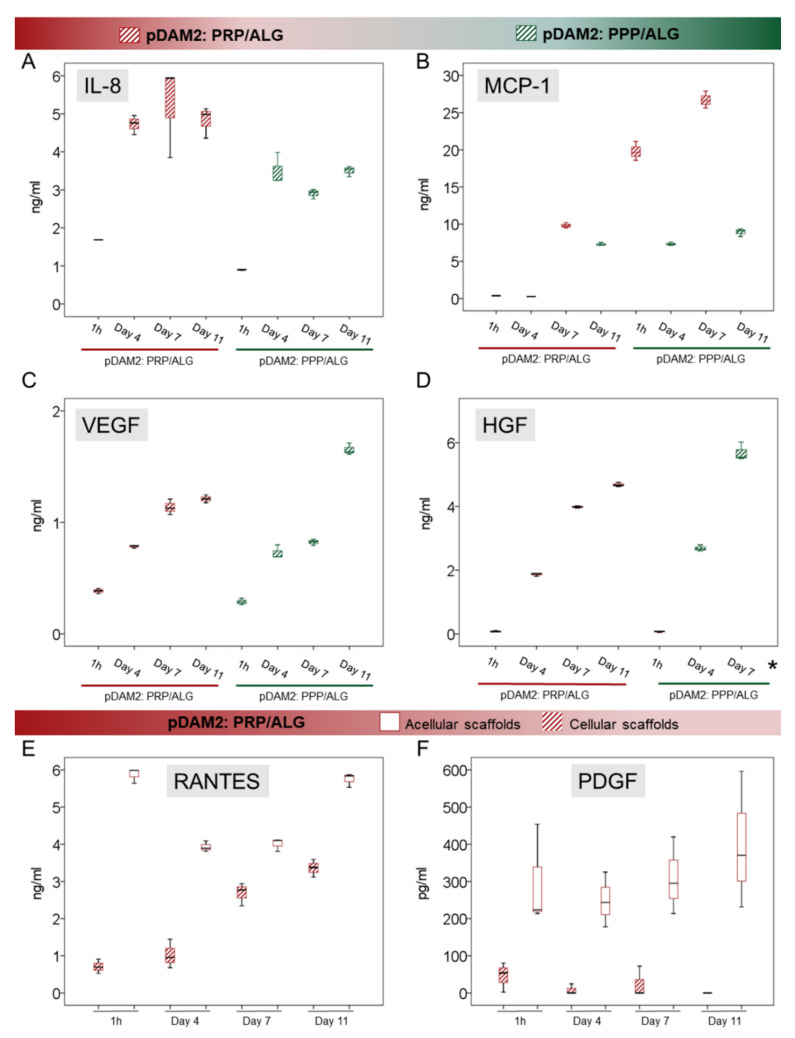
(**A**–**D**) Box plots show the median and 25–75 percentiles of relevant signaling cytokines involved in the modulation of inflammation (IL-8 and MCP-1, (**A**) and (**B**), respectively) and in angiogenesis (VEGF and HGF, (**C**) and (**D**), respectively) measured by ELISA in the conditioned media harvested over time. * Mean HGF concentration at Day 11 was 98,198 ± 3815 ng/mL (not shown in the graph). (**E**,**F**) RANTES and PDGF-BB released from the PRP component in the pDAM2: PRP/ALG dressings (as shown by acellular scaffolds cultured in the same conditions) were used up by embedded cells.

**Figure 7 ijms-23-02836-f007:**
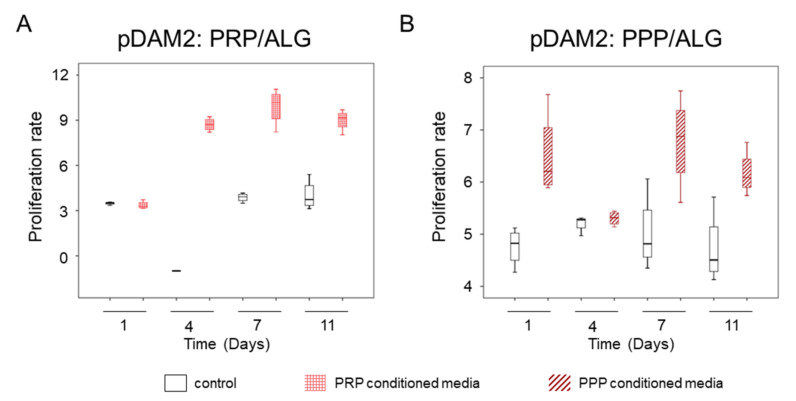
(**A**) Proliferative effect of conditioned media harvested from advanced dressings manufactured with PRP. (**B**) Proliferative effect of conditioned media harvested from advanced dressings manufactured with PPP.

**Figure 8 ijms-23-02836-f008:**
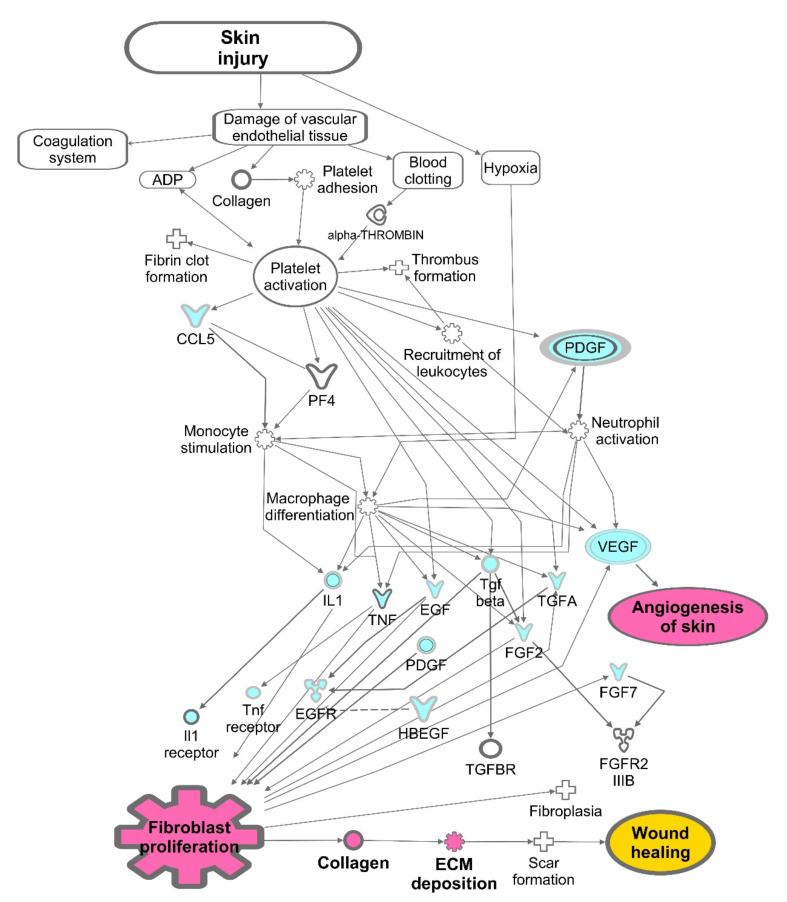
Illustration of molecules (blue) and mechanisms (pink) involved in the wound-healing cascade and assessed in this study.

## Data Availability

The data presented in this study are available upon request from the corresponding author.

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
