# Peer review of "Wound-Microenvironment Engineering through Advanced-Dressing Bioprinting"

_ijms, 2022, doi:10.3390/ijms23052836_

Round 1

Reviewer 1 Report

In this work, the authors reported the strategy of wound microenvironment engineering through 3D bioprinted advanced dressing for enhanced wound healing. The authors combined DAM, plasma and dermal fibroblasts to create bioinks for advanced dressing manufacturing. Two different modalities of composite bioinks, pDAM2: PRP/ALG and pDAM2: PPP/ALG, were prepared and compared. The multiplexing analyses of signaling proteins were helpful to predict and understand the activities in the wound environment. However, there are some issues should be addressed.

  1. The results have included the cell viability, proliferation, and protein expression, etc. in the wound microenvironment, indicating the effects of advanced dressing. However, there was no direct results of wound healing such as wound closure rates. The connection between wound microenvironment and wound healing has not been proved with strong support.
  2. The authors provided SEM images of the ink and its components. However, there was no discussion about these images. How would the morphology shown in SEM influence the ink performances or bioprinting process?
  3. Although conclusion section is not mandatory, it is highly suggested to add this part to summarize the long discussion section and highlight the significance of this research clearly.

Author Response

Many thanks for your comments that have helped us to improve our manuscript. We have answer one-by-one your concerns and modified the manuscript accordingly

In this work, the authors reported the strategy of wound microenvironment engineering through 3D bioprinted advanced dressing for enhanced wound healing. The authors combined DAM, plasma and dermal fibroblasts to create bioinks for advanced dressing manufacturing. Two different modalities of composite bioinks, pDAM2: PRP/ALG and pDAM2: PPP/ALG, were prepared and compared. The multiplexing analyses of signaling proteins were helpful to predict and understand the activities in the wound environment. However, there are some issues should be addressed.

  1. The results have included the cell viability, proliferation, and protein expression, etc. in the wound microenvironment, indicating the effects of advanced dressing. However, there was no direct results of wound healing such as wound closure rates. The connection between wound microenvironment and wound healing has not been proved with strong support.

We have further emphasized this limitation (page 14, lines 367-372. This is a preliminary work, which uncovers inflammatory modulation (e.g. IL-8, MCP-1) and potential promotion of angiogenesis (e.g. VEGF, HGF); here we show that these advanced dressings act as a dynamic source of functional proteins involved in angiogenesis and inflammation. Further in vivo studies analyzing their behavior in a proinflammatory and/or ischemic environment are needed to validate these composite bioinks and the manufacturing procedure in wound healing.

We have applied for further financial aid to validate these data in vivo, in both inflammatory and ischemic wounds.

  1. The authors provided SEM images of the ink and its components. However, there was no discussion about these images. How would the morphology shown in SEM influence the ink performances or bioprinting process?

Thank you for your comment.

We have discussed SEM images and bioprinting advantages of the blend in page 5, lines 158-159

Scanning electron microscopy of the pDAM2 hydrogels showed a randomly oriented fibrillar structure, with an average fibre diameter less than 100 nm and interconnecting pores. PRP/ALG and PPP/ALG hydrogels exhibited microstructure with many irregular aggregates. However, the images of pDAM2:PRP/ALG and pDAM2:PPP/ALG hybrid hydrogels revealed that those aggregates were covered by the fibrillar structure of the pDAM2 hydrogels.

Regarding the advantages in bioprinting process, the behaviour of the hydrogel blends (pDAM2: PRP/ALG and pDAM2: PPP/ALG) was superior to the plasma bioinks individually and allowed a reduction of the diameter of the extrusion needle (from 20G to 22G) improving filament homogeneity, without detrimental consequences in cell viability

  1. Although conclusion section is not mandatory, it is highly suggested to add this part to summarize the long discussion section and highlight the significance of this research clearly.

Many thanks for your comment. A conclusion section has been added in page 19, lines 552-559.

Reviewer 2 Report

I reviewed the manuscript proposed by Cristina Del Amo and co-workers titled
“Wound microenvironment engineering through advanced dressing bioprinting”.

The novelty of bioink composition and preparation is well defined.
The research methods presented in the manuscript are of good quality.
After revision the manuscript can be published nicely and is a valuable
contribution to the scientific community.

The introduction covers the aim of the study with sufficient citations.
Description about the use of Platelet Rich Plasma and adipose matrix can
be improved by adding more details as a function of previously published
literature.

Why is 3D printing relevant for skin repair purposes? Please highlight the advantages of 3D printing technique related to this application. Pag. 3 Line 119. What the Authors mean for “proper 3D printing resolution”?

The bioinks here presented can be reproduced in an easy manner?

Is the composition of PRP main components known?

Is the composition of pDAM2 known?

Pag. 6 Line 152. Is it correct to normalize protein concentration over time with respect to protein data at 1h? What about normalization with respect to the total protein concentration?

The use of hydrogels/bioinks containing proteins from PRP can alter the ELISA quantitation of the proteins expressed by cells over time due to bioink protein release?

The quality of Figure 1 and 2 can be improved.

Author Response

Many thanks for your comments that have helped us to improve our manuscript. We have answered one-by-one your concerns and modified the manuscript accordingly

I reviewed the manuscript proposed by Cristina Del Amo and co-workers titled 
“Wound microenvironment engineering through advanced dressing bioprinting”.

The novelty of bioink composition and preparation is well defined. 
The research methods presented in the manuscript are of good quality. 
After revision the manuscript can be published nicely and is a valuable 
contribution to the scientific community.

The introduction covers the aim of the study with sufficient citations. 
Description about the use of Platelet Rich Plasma and adipose matrix can be improved by adding more details as a function of previously published literature.

We have added more details of the importance of combining PRP and adipose matrix, in both research and clinical contexts. Page 2 lines 91-96, references [20, 21, 22]

Why is 3D printing relevant for skin repair purposes? Please highlight the advantages of 3D printing technique related to this application.

Thank you for your comment. We have added a paragraph, page 2 lines 60-64, highlighting the advantages of bioprinting related to this application

Pag. 3 Line 119. What the Authors mean for “proper 3D printing resolution”?

We have added an explanation, in relation to “proper 3D printing resolution” Page 5 lines 158-159, we refer to one of the most relevant parameters to consider during the extrusion bioprinting: the printing accuracy, which comprises the uniformity of the filament and the shape fidelity, as it has been reported in previous work (Del Amo et al. 2021).

The bioinks here presented can be reproduced in an easy manner?

It is relatively easy. The protocol for decellularization and elaboration of pDAM2 from adipose tissue is reported in a previous article by our group, please see reference [17].

The elaboration and the study of printability of plasma bioinks has been published previously by our group, please see reference [20]. Here we describe the combination of both.

Is the composition of PRP main components known?

Yes, the platelet secretome is complex and contains a large pool of signaling proteins. Because of their relevance in healing mechanisms, they have been intensely investigated in the last years. We have performed a significant contribution to unraveling the signaling secretome of platelets, please see reference 50

Is the composition of pDAM2 known?

Yes, it has been reported previously, reference [17]. We have added more information in the introduction, page 2 lines 79-82

Pag. 6 Line 152. Is it correct to normalize protein concentration over time with respect to protein data at 1h? What about normalization with respect to the total protein concentration?

Total protein concentration could be assessed (e.g. BCA), but it represents bulk proteins such as albumins fibronectin etc, which are very abundant (mg/mL) but not representative in cell signaling research. To obtain unambiguous information normalization should be performed in a one-by-one protein basis Careful normalization of each signaling protein with the value it had at the beginning of the culture period provides precision for unmistakable description of cell activities.

The use of hydrogels/bioinks containing proteins from PRP can alter the ELISA quantitation of the proteins expressed by cells over time due to bioink protein release?

Thank you for your comment. Cellular and acellular constructs were 3D printed and cultured overtime, in parallel. To confirm de novo synthesis of signaling proteins by embedded fibroblasts we have quantified the signaling proteins released from acellular scaffolds overtime and subtracted from the protein released in cellular scaffolds. This issue has been better clarified in the Methods section, page 17, lines 530-531.

The quality of Figure 1 and 2 can be improved.

We have improved the quality of the Figures, as suggested.

Round 2

Reviewer 2 Report

All the comments were properly addressed and the paper can be published in the revised form.